ecology/health and disease and epidemiology

food insecurity, insurance hypothesis, somatic maintenance, telomeres, birds, starlings

**Author for correspondence:**
Daniel Nettle
e-mail: daniel.nettle@ncl.ac.uk

# Exposure to food insecurity increases energy storage and reduces somatic maintenance in European starlings (*Sturnus vulgaris*)

Clare Andrews[1], Erica Zuidersma[2], Simon Verhulst[2], Daniel Nettle[3] and Melissa Bateson[4]

[1]Department of Psychology, University of Stirling, Stirling, UK
[2]Groningen Institute for Evolutionary Life Sciences, University of Groningen, Groningen, The Netherlands
[3]Newcastle University Population Health Sciences Institute, and [4]Newcastle University Biosciences Institute, Newcastle University, Newcastle, UK

SV, 0000-0002-1143-6868; DN, 0000-0001-9089-2599; MB, 0000-0002-0861-0191

Birds exposed to food insecurity—defined as temporally variable access to food—respond adaptively by storing more energy. To do this, they may reduce energy allocation to other functions such as somatic maintenance and repair. To investigate this trade-off, we exposed juvenile European starlings (*Sturnus vulgaris*, $n = 69$) to 19 weeks of either uninterrupted food availability or a regime where food was unpredictably unavailable for a 5-h period on 5 days each week. Our measures of energy storage were mass and fat scores. Our measures of somatic maintenance were the growth rate of a plucked feather, and erythrocyte telomere length (TL), measured by analysis of the terminal restriction fragment. The insecure birds were heavier than the controls, by an amount that varied over time. They also had higher fat scores. We found no evidence that they consumed more food overall, though our food consumption data were incomplete. Plucked feathers regrew more slowly in the insecure birds. TL was reduced in the insecure birds, specifically, in the longer percentiles of the within-individual TL distribution. We conclude that increased energy storage in response to food insecurity is achieved at the expense of investment in somatic maintenance and repair.

# 1. Introduction

When birds such as starlings are exposed to food insecurity—defined as temporally variable access to food—they respond by storing fat and gaining body mass [1–6]. This is an adaptive response: the greater the risk of a period of the shortfall, the larger the energy buffer it is optimal to store [7–10]. Though increased fat storage in response to food insecurity has been best studied in small birds, it may be more broadly distributed. It has been demonstrated in rodents [11–13]. In humans, the experience of food insecurity, as measured by questionnaire, is associated with a higher body mass index in women but not in men [8,14]. It has been widely assumed that the mechanism underlying food insecurity-driven mass gain is increased food consumption during the times when food is available [15–17]. However, the empirical evidence does not currently support this assumption. In food insecurity experiments, birds can gain weight while not increasing their food consumption, or even while decreasing it [3,5,6,18]. Likewise, food-insecure women have higher body mass indices without apparently consuming any more calories [19–23]. Another possibility is that food-insecure individuals sequester more energy for fat storage by reducing their energy expenditure rather than increasing their intake. In related work, Wiersma & Verhulst [24] showed that when foraging was made more costly by mixing food with chaff, zebra finches (*Taeniopygia guttata*) decreased their daily energy expenditure, despite the greater time spent foraging.

There are several ways that an animal might reduce energy expenditure. Zebra finches have been shown to reduce activity in response to food insecurity [25], though recent evidence from European starlings was inconclusive [5]. Beyond physical activity, animals may downregulate investments in somatic maintenance and repair. In zebra finches, Marasco *et al.* [26] found that long-term exposure to food insecurity increased the rate of accumulation of DNA damage (as measured by 8-hydroxy-2′-deoxyguanosine). Wiersma & Verhulst [24] found that zebra finches whose foraging costs were increased regrew a plucked tail feather more slowly than control birds. Another possible marker of somatic maintenance is telomere length (TL). Telomeres are repetitive DNA sequences at the ends of chromosomes that serve to maintain chromosome integrity [27]. They gradually shorten with age due to end-replication problems and other processes [28], shortening that is accelerated by oxidative stress (review; [29]). Across non-human vertebrates, shorter TL or accelerated telomere shortening is associated with ecological challenges such as infection, high competition, poor food or harsh abiotic conditions (meta-analysis; [30]). In nestling European starlings (*Sturnus vulgaris*), nutritional shortfall and increased begging effort accelerate telomere shortening [31]. Individuals can invest in maintaining TL through antioxidant defences [32]. Thus, change in TL in a proliferative tissue such as blood can be used as an index of investment in somatic maintenance and repair. In zebra finches, TL measured in erythrocytes is well correlated with TL in other tissues [33].

In the present study, we exposed groups of captive wild-caught juvenile European starlings to an extended period (19 weeks) of either food insecurity or constant food availability. Our method of imposing food insecurity was similar to that of several earlier studies [2,26,34]: the removal of access to any food for a 5-h period in the 15-h day, whose timing during the day varied randomly. In the present case, this was done 5 days out of 7, with uninterrupted access to food on the remaining 2 days. Note that this manipulation introduces both restriction of food access, and temporal unpredictability, to the insecure birds compared to the controls. It was not the aim of this study to distinguish the effect of unpredictability from that of restriction, as some other studies have done [1,3]. Food insecurity in the wild may typically involve both, and we were simply seeking a food insecurity regime sufficient to affect the birds in a naturalistic manner. We measured body mass repeatedly, as well as fat scores at the end of the treatment period. In addition, we measured two potential markers of somatic maintenance and repair, induced feather regrowth and erythrocyte TL. We measured TL by terminal telomere restriction fragment analysis. This has the advantages, compared to the popular qPCR relative TL assay [35], of higher precision, and providing, for each sample, a distribution of the lengths of the telomeres present, not just a single estimate of central tendency [36]. We also gathered some information on food consumption, though for logistical reasons the consumption data did not cover every day of the study period. Our general hypothesis was that food insecurity would produce an increase in energy storage and decreased expenditure on somatic maintenance and repair. Hence, we predicted body fat and mass would increase, while the rate of feather regrowth and TL would decrease, under food insecurity compared to the control treatment.

## 2. Methods

### 2.1. Birds and aviaries

We captured and retained 70 European starlings from a baited site in Northumberland over 4 days in October 2016, using a whoosh net. The number of birds was limited by aviary capacity constraints, but was several-fold larger than the numbers of animals used in comparable previous experiments (typically 6–24; [3–6,24,37]). Juvenile status (having hatched in Spring 2016) was still recognizable from plumage and only juveniles were retained. We focused on juveniles in order to standardize the ages of our sample, and also because TL changes more rapidly earlier in life [38]. Birds were transported in cloth bags to the laboratory (approx. 30 min). On arrival, birds were weighed and tarsus length measured (see body mass, fat scores and tarsus length, below). Sex was established from visual appearance, principally iris colour; sexing this way has been shown to be highly reliable in starlings [39]. An initial blood sample was taken (see blood sampling, below); one tail feather pulled (see feather regrowth, below); and a numbered plastic leg ring fitted. Birds were also treated with topical Ivermectin to kill common parasites.

Birds were then released into one of four indoor aviaries, where they remained for the duration of the experiment. The aviaries varied slightly in size, with width 239–246 cm, depth 209–219 cm and height 240 cm. The light cycle of the aviaries was 15 L : 8 D with dim lighting simulating dawn/dusk during the first/last 30 min of the light period. Drinking water was available at all times and environmental enrichment was provided in the form of rope perches, water baths and wood-shaving substrate. Diet throughout the study was a mixture of commercially available dry cat food (Royal Canin Ltd.), chick crumb (Special Diets Services 'Poultry Starter (HPS)') and insect mix for birds (Orlux insect patee). Birds were left to settle in their aviaries with ad libitum food for 11–19 days prior to the beginning of the experimental treatment.

One bird was euthanized prior to the beginning of the treatment, owing to lethargy and very low body weight. This left a final sample of 69 birds, assessed as 29 females and 40 males. On conclusion of the experiment, birds were given a period of ad libitum food, inspected by a veterinarian, transported to the site of capture in cloth bags and released.

### 2.2. Experimental treatments

Two aviaries each were assigned to the two experimental treatments (insecure and control). The assignment was by alternation within sex, on removal from the bags, and so was effectively random apart from sex balancing. This produced 35 birds (20 male) in the insecure treatment and 34 (20 male) in the control treatment. For 5 days a week (Monday to Friday), food was provided in automated pet feeders (Andrew James Ltd; three per aviary). These worked by sequentially revealing each of four compartments at pre-programmed times (i.e. one compartment was open at any one time). For the control treatment, all compartments were full of food. Thus, although each compartment was revealed at the same time as for the insecure treatment, food was always available. For the insecure treatment, one compartment was empty, and thus no food was available for 5 h out of the day. The time of onset of the period without food was varied pseudo-randomly from day to day by re-programming the feeders. It was the same for the two aviaries in the insecure treatment on any given day. Food deprivation could begin at any hour within the period of full light and could last until dusk (i.e. the earliest onset of the 5 h period was after the 30 min of dawn, and the latest end of the 5 h period was at the beginning of the 30 min of dusk). The amount of food in each non-empty compartment was sufficient that food never ran out outside the intended 5 h period of deprivation. On the remaining 2 days of the week, uninterrupted food access was provided to both aviaries all day in open bowls. During week 9, uninterrupted food access was provided to both groups every day, as the facility was closed for a public holiday. The experimental treatment was continued for a total of 19 weeks.

### 2.3. Body mass, fat scoring and tarsus length

Body mass was measured on arrival, immediately prior to the beginning of the experimental treatment (henceforth baseline), then after 2, 5, 8, 11, 14, 17 and 19 weeks of treatment. Other than the day of arrival, birds were always weighed before dawn. Each bird was placed in a plastic cone on a digital scale measuring to a resolution of 0.1 g.

In addition, all birds were manually fat scored at week 19 (0–8, Biometrics Working Group system [40]) by CA, who was not blind to treatment. Fat score was positively correlated with mass ($r = 0.58$, $p < 0.001$). A subset of 14 birds was also fat scored independently by a different, experienced avian fat scorer blind to treatment. The repeatability (intra-class correlation coefficient, ICC) for the two raters was 0.75 (95% CI 0.72–079).

Tarsus length, a measure of skeletal size, was also taken on the day of capture. The right tarsus was measured twice independently, using digital callipers with a resolution of 0.1 mm. The ICC for the two measurements was 0.83 (95% CI 0.76–0.88). The average of the two measurements was used for analysis.

## 2.4. Food consumption

Food consumption was estimated for 4 days out of every 7 by weighing the food remaining in the automated feeders. Due to logistical constraints, it was not possible to weigh the food on Fridays, Saturdays or Sundays. Thus, the food consumption data are incomplete and do not cover the 2 days per week when the insecure birds had ad libitum food. Food consumption was only measured at the aviary level. Food weighings were not blind to treatment. We averaged across the 4 days of each week to produce one consumption number for each aviary in each week and converted this to g per bird per day to correct for the different numbers of birds in each aviary.

## 2.5. Feather regrowth

On capture, we removed the left outer retrix (tail feather) by grasping the rachis with blunt-ended forceps and gently pulling until the feather was released. This was repeated after 5 and 17 weeks of treatment, by which times the pulled feather had largely grown back. The length of the regrowing feather was measured in mm using digital callipers (resolution 0.1 mm), from the base of the pin to the most distal point of the feather tip, after 2, 5, 8, 11, 14, 17 and 19 weeks of treatment. These measurements were blind to treatment. At week 17, three birds had feather lengths substantially shorter than they had been at week 14. These were assumed to represent breakage or accidental loss and excluded from the analysis.

## 2.6. Telomere length

TL was measured in erythrocytes by telomere restriction fragment analysis under non-denaturing conditions. Blood samples (around 140 µl) were taken by puncture of an alar vein with a 25-gauge needle and collection into capillary tubes. Samples were transferred to EDTA-treated plastic tubes on ice. They were then centrifuged to separate cells from plasma (10 min at RCF 1400 g), and pellets of cells were frozen to −80°C. Blood samples were taken on arrival (henceforth baseline; note that this is two weeks earlier than the baseline date for mass), and after 2, 8, 14 and 19 weeks of treatment.

TL analysis followed the methods of Bauch *et al.* [41]. In brief, we washed the cells and isolated DNA from 5 µl of erythrocytes using CHEF Genomic DNA Plug kit (Bio-Rad, Hercules, CA, USA). Cells in the agarose plugs were digested overnight with Proteinase K at 50°C. Half of a plug per sample was restricted simultaneously with HindIII (60 U), HinfI (30 U) and MspI (60 U) for approximately 18 h in NEB2 buffer (New England Biolabs Inc., Beverly, MA, USA). The restricted DNA was then separated by pulsed-field gel electrophoresis in a 0.8% agarose gel (Pulsed Field Certified Agarose, Bio-Rad) at 14°C for 24 h, 3.5 V cm$^{-1}$, initial switch time 0.5 s, final switch time 7.0 s. For size calibration, we added 32P-labelled size ladders (DNA Molecular Weight Marker XV, Roche Diagnostics, Basel, Switzerland; NEB MidRange PFG Marker I, New England Biolabs, range 15–242.5 kb). Gels were dried (gel dryer, Bio-Rad, model 538) at room temperature and hybridized overnight at 37°C with a 32P-end-labelled oligonucleotide (5′-CCCTAA-3′)4 that binds to the single-strand overhang of telomeres of non-denatured DNA. Subsequently, unbound oligonucleotides were removed by washing the gel for 30 min at 37°C with 0.25× saline-sodium citrate buffer. The radioactive signal of the sample-specific TL distribution was detected by a phosphor screen (MS, Perkin-Elmer Inc., Waltham, MA, USA), exposed overnight and visualized using a phosphor imager (Cyclone Storage Phosphor System, Perkin-Elmer Inc.).

TL distributions were quantified using IMAGEJ (v. 1.38×). The TL parameters potentially relevant to ageing and somatic state are not just average TL, but aspects of an individual's TL distribution (for example, the length of the shortest or longest telomeres). We, therefore, calculated the mean of the TL distribution (i.e. henceforth aTL), and additionally the percentiles, in 5% intervals, from 10% to 90%. For each sample, the limit at the side of the short telomeres of the distribution was lane-specifically set at the point of the lowest signal (i.e. background

intensity). The limit on the side of the long telomeres of the distribution was set lane-specifically where the signal dropped below Y, where Y is the sum of the background intensity plus 10% of the difference between peak intensity and background intensity. The coefficient of variation of a control sample run on 15 gels was 6%. The ICC across individuals was 0.77, including treatment week as a fixed factor. Since true TL was presumably changing over the course of the study, 0.77 represents a minimum estimate of the technical repeatability of the TL measurements and suggests measurement precision was acceptably high [42]. All telomere measurements were conducted blind to treatment.

## 2.7. Statistical analysis

Data were analysed in R, v. 4.1.1 [43], using linear mixed models with R packages 'lme4' and 'lmerTest'. Model estimation used restricted maximum likelihood. Significance testing used Satterthwaite's method with $\alpha = 0.05$. We fitted separate models for each outcome variable: mass, fat score, food consumed, TL and feather regrowth. We used insecurity status as the fixed predictor, where this status was a control for all birds at baseline, and insecure for the insecurity groups subsequent to the onset of the experimental treatment. Taking the data from the onset of the experimental treatments onwards and using the treatment group as the fixed predictor produces very similar results. Preliminary inspection revealed week-to-week changes with no linear trend, especially for mass (perhaps due to temperature and seasonal variation). We, therefore, included treatment week as a fixed factor rather than a continuous covariate. The interaction between treatment week and insecurity was also included. The distributions of residuals were checked and found satisfactory for the assumptions of the models. For binary comparisons of means, we report Cohen's $d$ as measures of effect size.

Models for mass, TL and feather regrowth included random effects of birds to account for repeated measures. Adding aviary as an additional level of random effect did not improve AIC or change results, and hence was not included in the analyses presented below. In the models for mass, we included tarsus length as a covariate, to account for skeletally larger birds also being heavier. For TL, in the first model, the outcome variable was aTL. In a follow-up model, we used all available percentiles of the TL distribution. For this model, the fixed predictors were insecurity, week, percentile and all possible interactions, with random effects of sample identity and bird.

We had two measures of energy storage (mass and fat score), and the two principal measures of somatic investment (average TL and feather regrowth). An important inferential question is whether there is evidence of an experimental effect on energy storage from the former two measures combined, and on somatic investment from the latter two measures combined. Multivariate analysis would usually be used to address this type of question. However, in this case, the variable structure of the data made multivariate analysis impossible (for example, fat score was measured only once per bird, but mass was measured multiple times). For overall tests of treatment effects on energy storage and somatic investment, we, therefore, relied on two methods. First, we performed combined significance tests on each pair of measures, using the inverse normal method of combining multiple $p$-values [44] (implemented in R package 'metap'). Second, we combined the effect sizes and their confidence intervals from the two measures of the pair meta-analytically to obtain a pooled effect size (using R package 'metafor'). For the models feeding into the meta-analyses, dependent variables were standardized prior to analysis, and interactions between insecurity and week were excluded.

From our prior work [5], we did not expect sex-specific responses to food insecurity in starlings. However, the present experiment was well powered to test for these. We repeated all analyses additionally including sex, and all its interactions with treatment, as an additional factor. No interactions involving sex were significant, and the only significant main effect of sex was on body mass (as expected, males were consistently heavier than females, $F_{1,67} = 18.26$, $p < 0.001$; estimated marginal means: male 79.1 g, s.e. 0.59; female 75.2 g, s.e. 0.70). Other effects were not substantially altered by the inclusion of sex. For simplicity, we therefore report results below from the models without sex.

# 3. Results

## 3.1. Mass and fat scores

Mass at baseline did not differ significantly between the treatment groups (control: mean 75.20 g, s.e. 0.72; insecure: 74.80 g, s.e. 0.76; $F_{1,66} = 0.04$, $p = 0.85$). Figure 1$a$ shows mean mass by treatment group across time. The main effect of insecurity was marginally non-significant ($F_{1,448.33} = 2.78$, $p = 0.10$), but

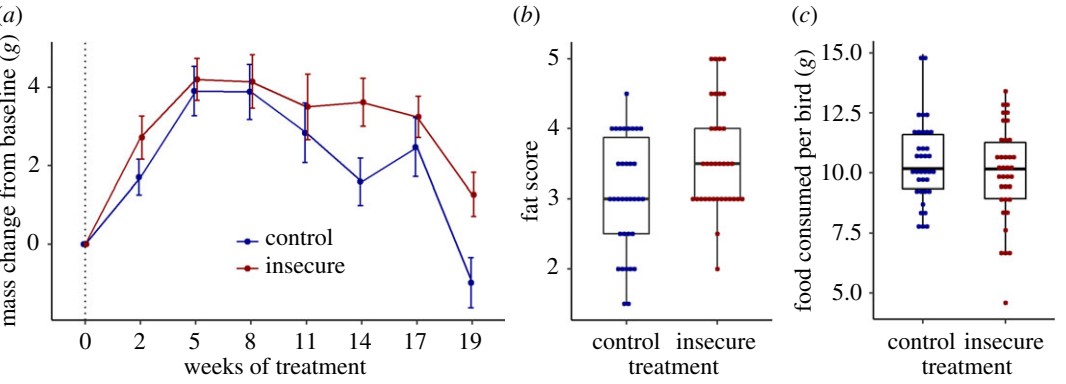

**Figure 1.** Effects of experimental treatment on mass, fat and food consumption. (*a*) Mass change from baseline ±1 s.e., by treatment across the experimental period. (*b*) Fat scores after 19 weeks of treatment, by treatment. Points represent birds, and boxes represent the median and lower and upper quartiles. (*c*) Food consumption (gram per bird per day), by treatment. Points represent aviary weeks, and boxes represent the median and lower and upper quartiles.

**Table 1.** Correlations across individuals for average TL at different time points.

|          | week 2 | week 8 | week 14 | week 19 |
|----------|--------|--------|---------|---------|
| baseline | 0.76   | 0.75   | 0.77    | 0.87    |
| week 2   |        | 0.76   | 0.77    | 0.75    |
| week 8   |        |        | 0.80    | 0.80    |
| week 14  |        |        |         | 0.85    |

there was a significant interaction between insecurity and week ($F_{6,469.38} = 2.23$, $p = 0.04$). The mass difference by insecurity status was substantial at weeks 14 (1.76 g; $d = 0.40$, 95% CI −0.08 to 0.89) and 19 (2.08 g; $d = 0.48$, 95% CI −0.01 to 0.97) and negligible at, weeks 5 (0.13 g; $d = 0.03$, 95% CI −0.45 to 0.51) and 8 (0.10 g; $d = 0.02$, 95% CI −0.46 to 0.50).

Fat scores at week 19 were significantly higher for the insecure group (mean 3.59, s.e. 0.13) than the control group (mean 3.06, s.e. 0.14; $t = 2.76$, $p = 0.01$; $d = 0.66$, 95% CI 0.17–1.15; figure 1*b*).

## 3.2. Food consumption

We calculated food consumed per bird at the aviary level, as described in Methods (i.e. there was one data point per aviary per week). In a model with insecurity, week and their interactions as fixed predictors, the main effect of insecurity was not significant ($F_{1,2} = 0.22$, $p = 0.68$; estimated marginal means: insecure 10.0 g, s.e. 0.71, control 10.5 g, s.e. 0.71). The interaction between week and insecurity was marginally non-significant ($F_{17,34} = 1.91$, $p = 0.05$).

## 3.3. Telomere length

Mean aTL at baseline was 17 351 bp (s.d. 1032, range 15 110–20 179). Individual TL showed high degrees of consistency over time; for example, the correlation matrix of aTL across individuals at the various time points is shown in table 1. Correlations over time for percentiles of the TL distribution were similar. On average, individuals' aTL shortened by 142 bp (s.e. 67) between baseline and the final TL measurement point 21 weeks later ($t = −2.12$, $p = 0.04$). By treatment group, shortening was 209 bp (s.e. 66) for the insecure birds ($t = −3.15$, $p = 0.009$), and 77 bp (s.e. 115) for the control birds ($t = −0.67$, $p = 0.51$).

At baseline, aTL did not differ significantly between treatment groups (control: mean 17420, s.e. 179; insecure: mean 17 285, s.e. 187; $t = 0.52$, $p = 0.61$). In the model using aTL as the outcome variable, the main effect of insecurity was marginally non-significant ($F_{1,326.78} = 3.03$, $p = 0.08$). The interaction between insecurity and week was not significant ($F_{3,262.43} = 1.30$, $p = 0.27$). Figure 2*a* shows aTL by treatment group at each measurement point. The difference between the two groups was largest at week 2 (−516 bp; $d = −0.45$, 95% CI −0.96 to 0.06) and smallest at week 14 (−204 bp; $d = −0.17$, 95% CI −0.65 to 0.30).

We followed up this analysis with a model using the full range of percentiles of the TL distribution. The main effect of insecurity was not significant in this model ($F_{1,298.3} = 2.57$, $p = 0.11$), and neither was

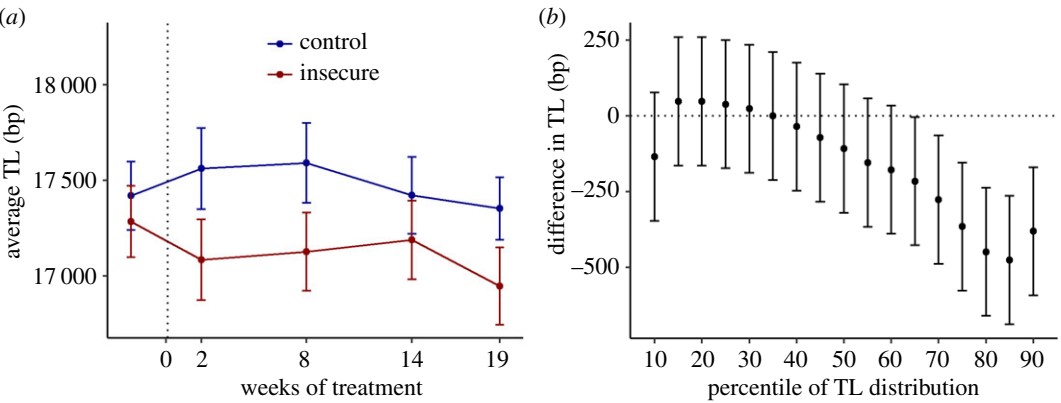

**Figure 2.** Effects of food insecurity on telomere length. (*a*) Average TL by insecurity status through the treatment. Error bars represent one standard error. The dotted vertical line represents the onset of the treatments. (*b*) Difference between insecure and control birds by percentile of the TL distribution collapsed across the weeks after the onset of the treatment. Data represent the difference in marginal means (±1 s.e.), estimated from the statistical model. A negative number indicates shorter TL in the insecure birds.

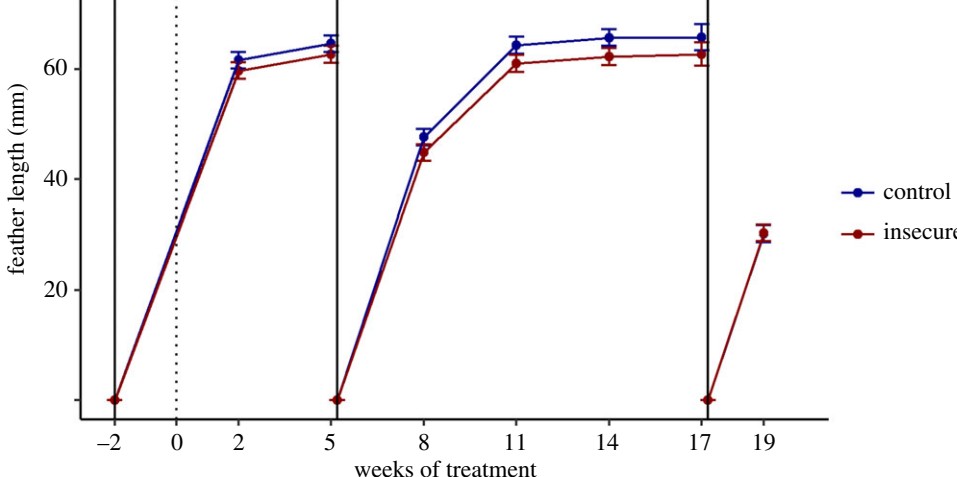

**Figure 3.** Length of regrowing tail feathers (mm) by insecurity and time point. The pulling of the feather is indicated by the vertical solid lines. The beginning of the treatment phase is shown with a vertical dotted line. Shown are estimated marginal means plus or minus one standard error. At weeks 17 and 19, the data are overlapping.

the main effect of week ($F_{4,244.2} = 1.53$, $p = 0.19$). There was, however, a significant interaction between insecurity and percentile ($F_{16,5257.6} = 11.04$, $p < 0.001$). No other interactions were significant. As figure 2*b* shows, the insecure birds had shorter TL at the longer percentiles of the TL distribution.

## 3.4. Feather regrowth

For feather regrowth, there was an expected large effect of week, because feathers grew from week to week and were re-pulled at weeks 5 and 17 ($F_{6,363.84} = 147.96$, $p < 0.001$). In addition, there was a significant effect of treatment ($F_{1,66.37} = 5.40$, $p = 0.02$). Birds from the insecure groups had slightly but consistently shorter feathers at all time points other than the final one (figure 3). The interaction between treatment and week was not significant ($F_{6,363.84} = 0.35$, $p = 0.91$). Averaging across the measurement points, insecure birds had an average feather length of 53.90 mm (s.e. 1.01) compared to 56.20 mm (s.e. 0.42) for the control birds, corresponding to an effect size (Cohen's *d*) of −0.50 (95% CI −0.01 to −0.99).

## 3.5. Combined *p*-values and meta-analyses

The combined effect of insecurity on the measures of energy storage, mass and fat score, was significant using the inverse normal combined *p*-value ($p = 0.004$). In a fixed-effects meta-analysis, there was a significant positive pooled effect of insecurity on mass and fat score (figure 4; $B = 0.25$, s.e. 0.09, 95%

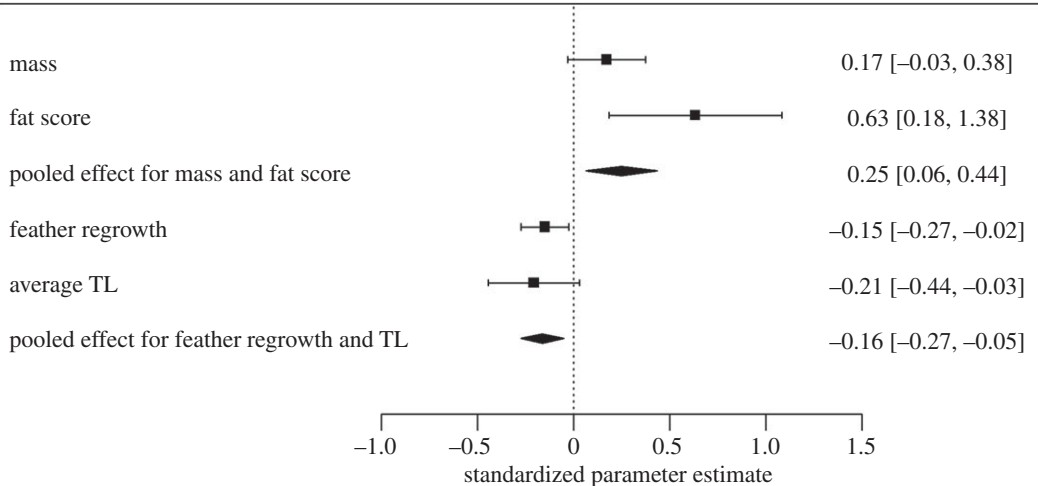

**Figure 4.** Meta-analysis of study measures. Squares represent standardized effect sizes, and whiskers represent 95% CIs. Diamonds represent pooled effect sizes and their 95% CI from a fixed-effects meta-analysis model.

$z = 2.64$, $p = 0.008$). The combined effect of insecurity on the measures of somatic investment, average TL and feather regrowth, was also significant ($p = 0.009$). In a meta-analysis, there was a significant negative pooled effect of insecurity on average TL and feather regrowth (figure 4; $B = -0.16$, s.e. 0.06, 95% $z = -2.86$, $p < 0.001$).

## 4. Discussion

We experimentally exposed groups of young starlings to food insecurity or uninterrupted food access over a period of more than four months. Overall, our results support the hypothesis that the birds experiencing food insecurity increased energy storage, and reduced somatic investment and repair. When the evidence from fat scores and masses was combined there was a clear pattern of increased energy storage, though the effect on mass considered separately was significant only in interaction with time point. On the somatic maintenance side, again the pattern of reduced investment was clearer when the evidence from TL and feather regrowth was combined. Considered separately, the effect of insecurity on TL was significant only in interaction with percentile of the TL distribution: the longer telomeres were those affected.

Our findings that food insecurity increased energy storage conceptually replicate earlier findings in starlings and other passerine birds [3–6,45]. The insecurity effect on mass varied from week to week; when averaged over all the weeks, the effect size was small. This is consistent with our recent findings from a series of experiments using a different method of inducing food insecurity in starlings. There, we either used 20-min blocks in which food was withdrawn or else made each attempt to forage from a feeder probabilistically unsuccessful. We found evidence for mass gain under food insecurity overall, but with effects that varied in magnitude from experiment to experiment and were null in some experiments [5]. How successful any of these laboratory protocols are at simulating natural food insecurity is not clear; it may be that they underestimate the magnitude or reliability of the shifts in the wild. For example, in the current experiment, birds would have been able to learn that the absence of food is always short-lived, something that is not necessarily true in the wild.

There are several non-mutually exclusive mechanisms that could explain how food insecurity induces fat storage. The first is that food-insecure birds consume more food in the periods when food is available [15–17]. We found no evidence for increased food consumption. This result is not definitive: in the present experiment, we only measured food consumption on 4 days out of every 7, and only at the coarse level of the whole aviary. Thus, we cannot exclude that the food-insecure aviaries consumed more food than the control aviaries on the 2 ad libitum days per week where food consumption was not monitored. Nonetheless, the non-significant trend we observed was in the direction of insecure birds eating less rather than more. This finding is consistent with a number of other avian studies where food consumption was measured more completely, in which food-insecure birds gained weight despite eating no more food, or less food [3,5,6,18]. It is also consistent with the human evidence that food-insecure women gain weight without apparently consuming any more calories, [19–23], though those studies suffer from the limitation that food consumption is self-reported.

A second possible mechanism is that food-insecure birds assimilate more of the potential caloric content of the food they do consume. In our previous study in the same species [5], we used bomb calorimetry to measure the energy density of guano. We found lower energy density of guano in food-insecure birds, suggesting greater assimilation of the caloric content. We did not collect guano in the present experiment, and hence have no information on whether assimilation was increased, though this is plausible given previous findings [5,46].

Third, food-insecure birds may reduce energy expenditure on other functions. Our findings on feather regrowth and TL suggest in particular that energy allocation to somatic maintenance and repair was reduced. These findings are consistent with Wiersma & Verhulst's [24] demonstration of reduced feather regrowth in zebra finches for whom foraging had been made less profitable, and the evidence from Marasco et al. [26], also in zebra finches, of a faster accumulation of DNA damage over time in birds exposed to a food insecurity regime very similar to the present one. Whereas our choice of energy storage measures was straightforward, in that mass and fat score are the directly relevant quantities, feather regrowth and TL are not the only possible measures of somatic investment. Previous studies suggest food insecurity may have similar negative effects on DNA damage, or immune function [18,26]. The fact that both our chosen measures showed some evidence of a reduction under food insecurity was either fortunate or suggests that reduction of investment under food insecurity is detectable across a range of possible markers of somatic investment. Such reduced investment would provide a general pathway to explain the reliable associations between food insecurity and subsequent poor health in humans [47,48]. It is, however, difficult to reconcile with findings that long-term exposure to a food insecurity regime increased life expectancy in zebra finches [49] unless food insecurity alters the outcome of a trade-off between survival and reproduction. We note also that we did not measure other components of energy expenditure, such as movement, thermoregulation [50], preparation for reproduction or song and song learning [51], that could have also been altered under food insecurity.

Our investigation of TL under food insecurity was notable for its high precision, compared to many other avian TL studies. This precision arose from measuring TL five times on the same individuals and using the terminal restriction fragment approach rather than the more widespread qPCR assay (see [36] for discussion of alternative TL measurement methods). This method has several advantages. First, as used here it excludes interstitial telomere sequences, which can be numerous and variable between individuals in birds. Thus, it provides a clean measure of terminal TL, which is the parameter of interest. Second, using this method we were able to characterize the absolute lengths, in base pairs, of telomeres in the European starling, whereas our previous work [52,53] reported only relative abundance of the telomeric sequence. The average TL for the whole sample, 17 351 bp, falls squarely within the range observed in birds, fairly similar to the values seen in blue tits (Cyanistes caeruleus) and zebra finches measured by the same method [54]. Third, measuring the terminal restriction fragment provides a distribution of TL for each sample. This revealed that, though the effect of food insecurity on average TL was non-significant, there was an interaction between insecurity and percentile of the TL distribution, with insecurity appearing to shorten the longest telomeres within individuals. In common terns (Sterna hirundo), Bauch et al. [55] found that the length of the longest telomeres was a better predictor than the average TL of survival and reproductive success. Bauch et al. suggest that this is due to the effects of environmental stressors being most visible in the longest percentiles of the TL distribution, where telomeres shorten fastest in absolute terms. Our findings represent a direct corroboration of this claim.

Our repeated measurement of TL also allowed us to characterize TL dynamics, albeit that the timescale was short for examining age-related shortening given the rate at which TL changes after early life. As in other studies using high-precision methods, TL was individually highly consistent over time, with those individuals with long average TL at the beginning of the study generally having long TL at the end [56]. Despite the restricted study period, we were able to observe TL shortening, though it was significant only in the insecure group. Averaging across the treatment groups suggests an annual shortening rate of around $350$ bp yr$^{-1}$ for juvenile starlings. This is in the range estimated for other passerine birds [57], albeit that our birds were less than six months old and likely to be losing TL faster than the whole-life rate.

Although we only studied TL in a single tissue, blood, our findings are likely to reflect telomere dynamics more generally. Mean TL varies across tissues, but individuals with short TL in blood tend to have relatively short TL in other tissues too [33,58,59]. Moreover, the rate of telomere shortening appears to be equivalent across tissues in adulthood [33,58,59], though this is not true in early life [60].

Our results suggest that when faced with food insecurity, starlings can respond adaptively by increasing energy allocated to fat storage, even without taking in any more food overall. At the same

time, they reduce allocation to somatic maintenance and repair. The costs of doing this are real and measurable, in terms of slowed feather regrowth and erosion of the longest telomeres. Over time, such reduced investments would presumably have measurable impacts on health. Classical theoretical work on optimal energy reserves treated increased predation risk as the fitness cost of fat storage [9,61,62]. The present work suggests that increased predation risk does not adequately capture all of the costs, since energy intake is limited, and the energy to fund storage must be diverted from other fitness-relevant functions.

Ethics. This study was completed under UK Home Office licence 70/8089 (licence holder Melissa Bateson) and with approval of the Animal Welfare Ethical Review Board at Newcastle University. Capture of birds from the wild was done with landowner permission under Natural England permit number 2016-57171-SCI-SCI. A copy of the ARRIVE guidelines 2.0 essential items check list [63] is included as electronic supplementary material.

Data accessibility. Code and raw data relating to this study are freely available in the Zenodo repository at https://zenodo.org/record/5036419 (doi:10.5281/zenodo.5036419).

Authors' contributions. Conceived the study, planned and performed the bird experiment: C.A., D.N. and M.B. Analysed data or contributed to data analysis: C.A., S.V., D.N. and M.B. Planned and performed telomere analysis: E.Z., S.V. Wrote the draft paper: C.A., D.N. and M.B. Edited and approved the paper: all authors.

Competing interests. We declare we have no competing interests.

Funding. This project has received funding from the European Research Council (ERC) under the European Union's Horizon 2020 research and innovation programme (grant agreement no. AdG 666669, COMSTAR).

Acknowledgements. We are grateful to Martin Hughes; Michelle Waddle and other staff members of the Comparative Biology Centre; and other members of the COMSTAR research group.

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
