## [Peer Review File · Royal Society Open Science]

Review History

RSOS-211099.R0 (Original submission)

Review form: Reviewer 1

Is the manuscript scientifically sound in its present form?

No

Are the interpretations and conclusions justified by the results?

Yes

Is the language acceptable?

Yes

Do you have any ethical concerns with this paper?

No

Have you any concerns about statistical analyses in this paper?

Yes

Recommendation?

Major revision is needed (please make suggestions in comments)

Comments to the Author(s)

This paper investigates the energy allocation trade-off between mass gain (fat storage) and somatic maintenance and repair under food insecurity. The authors performed a well-designed experiment where birds were exposed to a period of either food insecurity (food deprivation during 5 hours/day at randomly chosen times) or constant food supply. They combined measures of body mass and fat scores as indicators of energy storage with measures of telomere length and induced feather regrowth as potential markers of somatic maintenance and repair. Even under a relatively mild challenge and short time period, the authors were able to detect differences between the two experimental groups, which is highly suggestive. Although overall their results support the hypothesis that birds experiencing food insecurity increased energy storage and reduced investment in somatic maintenance/repair, the effects are somehow subtle. Indeed, the main effects of insecurity on body mass and TL (both aTL and the TL distribution) are non-significant, while the clearest effect is found on feather regrowth. Telomere shortening in the insecure birds was only evidenced in the longer percentiles of the TL distribution, which is on the other hand, an interesting result. However, this implies it is highly dependent on the method used (TRF vs. qPCR). In addition, some of the relationships are not fully convincing, such as those between food insecurity and fat scores and food consumption (as the authors acknowledged). Although the statistical approach is sound and sample sizes are adequate, some analyses should be revised to clarify their findings. Overall, their work is interesting and may have important implications as it points towards telomere dynamics as a candidate mechanism to underlie life-history trade-offs. In general, the paper is well-written. Nonetheless, I would like to see a more elaborated context showing current evidence on the topic across birds, and not only – or mostly – in the study species and humans.

Specific comments

L 35. I miss a broader general context in the first paragraphs of the introduction. For example, the first sentence could suggest that the generality of this phenomenon is limited to very few bird species. I would prefer to see how general this process is and what is the current evidence to support –or not– this response across taxa, or at least, across bird species. For example, is fat storage/mass gain more likely to occur in short-lived passerines than in long-lived large-sized birds (e.g. raptors, seabirds...)? And what is more, is increased energy storage at the expense of investment in somatic maintenance/repair equally beneficial/adaptive for short- and long-lived species? It would be interesting to develop these ideas in the introduction/discussion sections.

L 38-49. Here is the opposite. Although interesting, the jump from passerines (mostly starlings) to humans seems illogical, especially given the differences in methodology (in the case of humans, not only observational, but probably less accurate and highly biased data) and likely also in the drivers and underlying physiological mechanisms. It seems that you try to generalize patterns across vertebrates, but to what extent are they given the diversity of life-history strategies that exist in the animal kingdom? What is in between passerines and humans?

L 46. The qualifying term “relevant” is certainly unnecessary here. Please, remove.

L 48. Add the species’ Latin name

L 95. Please, indicate how many birds in total were captured.

L 96. Sex assignment through visual assessment seems to be highly reliable. Please, provide some details (or references) on the specific traits used to do this, as they may be very useful for other researchers working with the same study species.

L 112. There is an extra “prior”

L 140. Fat scoring of birds before starting the experiment would have been desirable. Do you have any reason for not having done so? Although both mass and fat storage are correlated, you cannot assume evidence for a treatment effect without having proof of change in this variable. It may also be other mechanisms underlying patterns of fat mobilization and storage (e.g sex-specific responses, diseases...).

In addition, measuring body size (not only body mass) would have been desirable, especially for the comparison of birds at the beginning of the experiment. In contrast to body mass, body size is not expected to change. However, the lack of differences in body mass could be related to birds' size and reporting this result without controlling for body size is not convincing. If data are available, I encourage authors to use it as a control variable.

L 149-150. I understand that measuring food consumption at the individual level under your experimental conditions is almost impossible, but the conversion to g per bird per day eliminates any type of variation between individuals, which is in considerable contrast to the other variables used in this study. I really appreciate the authors' transparency throughout the paper and this particular issue has been duly acknowledged.

L 157. Please, indicate the measurement error associated to your calipers.

L 196-97 Indicate whether CV values of the control sample and ICC across individuals are within the acceptable ranges. Surely they are, but please provide accepted values and references for that.

L 201. What is/are the response variable/s in the models of experimental effects? and what is the difference between these and models for mass, TL and feather regrowth, if any? Please, clarify

Sex should also be included in your models as a fixed factor. Sex-specific hormones, even in juvenile birds, may influence some physiological and immunological parameters, which may in turn influence trade-offs between different life-history traits. As an example, experimental implantation of testosterone in adult female starlings strongly suppressed tail feather regrowth after experimental plucking (De Ridder et al. 2002 *The Auk* 119). Even though sexes are quite balanced in your study design, you cannot completely rule out an effect of sex on your results without testing it.

L 217. Using a meta-analysis with data from a single study is uncommon and would need more justification. Please, explain why it is needed or what is the added value to your statistical approach? I guess it is not only a “life jacket” to reinforce otherwise weaker evidence... why not to do such an analysis with data from different studies?

L 228. This is a bit intriguing. Although both groups follow a quite similar trend, the mass difference by insecurity status at week 14 seems to be motivated by a substantial decrease in body mass of control birds from week 8th, which then recovers from week 14 to 17, and drops onwards. How do you explain these changes in body mass within the control group and, in particular, the sharp decrease at week 14 (when in addition, the difference in TL is the smallest)?

L 238. Please, indicate what the lines and whiskers in the box plots represent.

L 274. How do you explain the shorter TL in the insecure birds at percentile 10?

L 295. Confidence intervals for Mass and aTL contain zero. Although this does not certainly mean that there is no treatment effect, it does imply that it is uncertain whether there is a treatment effect. The combined effect for both mass and fat score clarifies this issue, but I have serious doubts about whether it is appropriate to include fat score in this analysis (for the reasons explained above). I suggest authors to re-run the analysis without this variable.

L 314-319 Why not to perform the meta-analysis with data from all these studies instead?

L 317. Indicate (briefly) what are the main differences between the two methods of inducing food insecurity, as these can influence the derived results and conclusions.

L 353-354. Although transparency and sincerity are highly appreciated, it is a bit strange to insist on this idea when the main hypothesis of this study includes a specific prediction about this component of the trade-off. This is not only because the TRF is a high-precision method and you have great expertise in telomere dynamics, which is great, but rather because telomere dynamics may be a candidate mechanism to underlie life-history trade-offs. Of course there are other measures that may also be involved in the trade-offs associated with energy limitations that are worth discussing. I suggest rewriting this paragraph to discuss in more detail how DNA damage, immune function, stress hormones and any other physiological trait may be affected by food insecurity, and how these plastic or adaptive complex responses may help organisms to cope with natural variation in environmental conditions, including food availability.

In addition, you may want to discuss the potential impact of other factors naturally affecting wild birds, such as infection by haemosporidians on your response variables. Your particular study species is not the best candidate, as they usually have very low prevalence of infection by these parasites, but considering this in future studies with other bird species would be important. Avian malaria infection may or may not have direct short-term costs (e.g. decreases in body mass) but may affect body condition and immune responses of birds (e.g., Valkiunas et al. 2006 *J. Parasitol.*, 92; Martínez-de la Puente et al. 2010 *Biol. Lett.*, 6; Marzal et al. 2008 *J. Evol. Biol.*, 21; Navarro et al. 2003 *Oikos*, 101) as well as susceptibility to oxidative stress (van de Crommenacker et al. 2002 *Proc. R. Soc. B* 279). On the other hand, diet quality / quantity may increase susceptibility to infectious diseases (Cornet et al. 2014 *J. Anim. Ecol.*, 83). All these effects may in turn influence telomere length and dynamics. For example, a causal relationship between avian malaria and faster telomere shortening has been demonstrated by the finding that telomeres shortened faster in experimentally infected captive birds compared with controls (Ashgar et al. 2015 *Science* 347). To add even more complexity, effects of infection by haemosporidians on telomere dynamics may be sex-specific (Sudyka et al. 2019 *The Science of Nature* 106). I believe it's worth considering these potential relationships in further studies.

L 362. Or the opposite! Shorter telomeres driven by poor environmental conditions –in a broad sense– could lead to faster pace-of-life, where investment in self-maintenance is reduced in order to save energy for reproduction, at the cost of somatic durability (see for instance Giradeau et al. 2019 *BioEssays*). You may want to further discuss these ideas and whether these patterns may also differ between individuals of different age or breeding prospects.

L 372 and 377. Please, provide Latin names.

L 386-391. Move this information to results.

L 392. I believe your results do not confirm, but rather suggest this may happen; evidence for fat scores and food intake is weak.

Hope you find these comments useful.

Review form: Reviewer 2

Is the manuscript scientifically sound in its present form?

Yes

Are the interpretations and conclusions justified by the results?

Yes

Is the language acceptable?

Yes

Do you have any ethical concerns with this paper?

No

Have you any concerns about statistical analyses in this paper?

No

Recommendation?

Accept with minor revision (please list in comments)

Comments to the Author(s)

This study aimed at understanding the effects of food insecurity on energy storage and somatic maintenance. To do so, the authors carried out a 19-week experiment on 69 European starlings kept on captivity. They predicted that insecurity would increase energy storage (here measured as mass and fat deposition) but, as a trade-off, decrease somatic maintenance (here measured as telomere length and feather growth). The goal of the study is very clear and the justifications for the hypothesis, the experimental design and the predictions explicit and concise. Globally, the results of the study go in the direction of the predictions even though the effect of insecurity on mass and average telomere length was significant at some time points only. Nonetheless, the results remain convincing. More importantly, the discussion clearly highlights the significant and non-significant effects of the treatment, and the conclusions are justified by the results. The authors also highlight the limitations of the study.

My main comment is about the result section. I find misleading the description of the differences between treatments of the mean values of the different traits. I suggest to only keep the interpretation of the outputs of the statistical models (see comments 9, 12, 13 and 14 below).

1. Line 95: The authors should explain why the study focusses on juveniles only. Was it to standardise the age, and, therefore, initial telomere length?
2. Line 109: I understand that the authors want to highlight in this paragraph how the study dealt with ethical aspects. However, this sentence looks a bit out of place. I would move this sentence line 136, when explaining how birds were weighed.
3. Line 112: Delete one of the "prior".

4. Line 122: I would precise between brackets that "one compartment was open every 5 hours".
5. Line 123: For clarity, I would rephrase as "Thus, although the compartments were revealed at the same times as for the insecure treatment, the compartments always contained food".
6. Line 127: This sentence suggests that the compartments do not open at fix interval; please clarify.
7. Line 130: This sentence suggests that birds always had access to food, even in the insecure treatment. This would mean that, contrary to what is written in the introduction, the manipulation did not introduce restriction of food access. Please clarify.
8. Line 136: I would move this sentence ("Body masses [...]") after "[...] 19 weeks of treatment." and remove the part about fat score as the fact that the experimenter was not blind to treatment when measuring fat deposition is written further in the paragraph.
9. Line 227: I would remove this sentence ("Insecure birds were [...]") because, as written after, insecure birds were significantly heavier than control birds at weeks 14 and 19 only.
10. Figure 1: I suggest adding letters to the 3 different graphs to highlight what are the significant differences.
11. Line 245: The model should be described in Statistical analyses.
12. Line 248: Because the effect of insecurity is not significant, this sentence is misleading. Alternatively, the authors may describe the marginally non-significant interaction between week and insecurity.
13. Line 263: Again, I think that interpreting the differences between the means (while ignoring the variance) is wrong. The interpretation should be only made from the output of the statistical models. This sentence says that insecure birds had shorter aTL than control birds while the previous sentence says that the effect of the treatment is marginally non-significant. Alternatively, the authors may write that "while the mean value of aTL was lower in the insecure group than in the control group, this difference was only marginally non-significant".
14. Line 265: Both CI include 0; if I am correct, this means that aTL did not differ between the two groups whenever it was at week 2 or 14.
15. Line 271: It would be better to interpret the interaction between insecurity and percentile from pairwise comparisons (using the "eemans" function for instance).
16. Figure 2: Figure 2A suggests a rather strong effect of insecurity on average telomere length. However, this effect is statistically weak. Therefore, I think the figure is misleading and the authors should represent aTL change from baseline rather than aTL (similarly to Figure 1A). The authors should also use letters or asterisks to highlight the differences that are significant.
17. Figure 2B: I find this figure very convincing. The interpretation of the interaction between insecurity and percentile should be emphasised using pairwise comparisons.
18. Line 282: It is probably worth repeating here when the feathers were plucked.

19. Lines 363-391: The two paragraphs about telomeres do not strictly relate to the study but provide interesting results that are valuable for research on telomeres in general.

Review form: Reviewer 3

Is the manuscript scientifically sound in its present form?

Yes

Are the interpretations and conclusions justified by the results?

Yes

Is the language acceptable?

Yes

Do you have any ethical concerns with this paper?

No

Have you any concerns about statistical analyses in this paper?

Yes

Recommendation?

Accept with minor revision (please list in comments)

Comments to the Author(s)

General comments:

Andrews and colleagues conducted a study to investigate how food insecurity affect energy storage and somatic maintenance in European starlings. The authors found that food insecurity resulted in increased body mass and fat score, as well as decreased telomere length and slower feather regeneration. This is a well-designed study and a well-written manuscript. I only have a few general comments and some minor suggestions to improve this manuscript. The manuscript would benefit from addition of a figure with experimental timeline, as well as pictures or videos of the automated feeder and different phases of feather regeneration after plucking. Furthermore, I suggest discussing more about the validity of erythrocyte telomere length (more below) and the role of telomerase in the context of food security and somatic maintenance. Specific comments are detailed below.

Specific comments:

Line 48: Please include Latin name of zebra finches.

Line 52: Please include Latin name of European starlings.

Line 64-66: Erythrocytes are terminally differentiated cells. They should not be proliferating.

Line 65: Are there any evidence that shows that TL in erythrocytes is representative of TL in other tissues?

Line 79-82: I think this belongs to the methods section.

Line 108: Is 11-19 days sufficient for acclimation to captivity? Did body mass change throughout this period?

Line 110: What does "birds' dawn" mean?

Line 121: A picture, or even better, video of how the automated feeders work would be helpful.

Line 198: The study includes both males and females but 'sex' was not included in the analysis. Was there a sex-specific effect and/or interaction?

Figure 1A: It seems like there is a significant drop in body mass in both groups but especially in the control birds from week 17 to week 19. Why?

Line 382-391: I think the measurement of TL in avian erythrocytes warrants a bit more discussion. How does this extrapolate to TL of other somatic tissues? Although the authors did not measure telomerase activity, the role of telomerase should be discussed briefly in the context of this study.

Decision letter (RSOS-211099.R0)

Dear Dr Nettle

On behalf of the Editors, we are pleased to inform you that your Manuscript RSOS-211099 "Exposure to food insecurity increases energy storage and reduces somatic maintenance in European starlings" has been accepted for publication in Royal Society Open Science subject to minor revision in accordance with the referees' reports. Please find the referees' comments along with any feedback from the Editors below my signature.

Please submit your revised manuscript and required files (see below) no later than 7 days from today's (ie 02-Aug-2021) date. Note: the ScholarOne system will 'lock' if submission of the revision is attempted 7 or more days after the deadline. If you do not think you will be able to meet this deadline please contact the editorial office immediately.

on behalf of Dr Kimberley Mathot (Associate Editor) and Kevin Padian (Subject Editor)
openscience@royalsociety.org

Associate Editor Comments to Author (Dr Kimberley Mathot):
Dear Dr. Nettle,

Thank you for submitting your manuscript for consideration at Royal Society Open Science. I have now received reports from three referees with relevant expertise, who were all generally

very positive about the work. the reviews were very constructive and point out several areas where the manuscript could be improved/clarified. Please provide a point-by-point response to all the referee comments, including the following points which provide my own view on some of the more substantial comments from the referees:

- 1) Both referee #1 and referee #3 raise questions about the possibility of sex specific effects. It seems reasonable to expect that males and females may respond to food insecurity differently. For example, loss of fat stores may impact female reproductive potential more than it would in males, and that females might be more willing to sacrifice somatic maintenance for fat stores. From your introduction (lines 38-39), it seems like this is the observed pattern in humans; in females, but not males, food insecurity is associated with higher body mass index. Perhaps my interpretation of this is incorrect, and the studies only involved female subjects, hence the qualifier. Could you please clarify whether these previous studies showed sex-specific effects, and if so, why sex specific effects weren't considered here? I believe with your sample sizes, you would have the power to address the interaction between food insecurity and sex.
- 2) Referee # 1 asks for clarification about the use of meta-analysis on multiple response variables from the same study and suggests the possibility of including previously published effect sizes in the current analysis. While I agree that such an expanded meta-analysis would be very timely and insightful, I think this is beyond the scope of the current work. Instead, can you please just provide a brief rationale of the approach you have used here.
- 3) Referee #2 had some concerns about the use of the term significant to describe effects throughout the manuscript when for certain specific time points, the effects were not statistically significant at a level of $\alpha \leq 0.05$. I believe this concern might be resolved by i) consistently relying on effect sizes and confidence intervals for drawing inference about effects and ii) providing an explicit statement about how you interpret support for a given effect (e.g., combination of estimated effect size and the likelihood that the effect is different from zero to infer strong, moderate, weak or no support). Referee #2 also felt that the results presented in Figure 2A were not consistent with the results from your statistical analyses, which gave a $p = 0.08$. To my eyes, they do appear consistent as the lack of difference in week 14 could be sufficient to make the overall effect "not significant" -but please double check.

Reviewer comments to Author:

Reviewer: 1

Comments to the Author(s)

This paper investigates the energy allocation trade-off between mass gain (fat storage) and somatic maintenance and repair under food insecurity. The authors performed a well-designed experiment where birds were exposed to a period of either food insecurity (food deprivation during 5 hours/day at randomly chosen times) or constant food supply. They combined measures of body mass and fat scores as indicators of energy storage with measures of telomere length and induced feather regrowth as potential markers of somatic maintenance and repair. Even under a relatively mild challenge and short time period, the authors were able to detect differences between the two experimental groups, which is highly suggestive. Although overall their results support the hypothesis that birds experiencing food insecurity increased energy storage and reduced investment in somatic maintenance/repair, the effects are somehow subtle. Indeed, the main effects of insecurity on body mass and TL (both aTL and the TL distribution) are non-significant, while the clearest effect is found on feather regrowth. Telomere shortening in the insecure birds was only evidenced in the longer percentiles of the TL distribution, which is on the other hand, an interesting result. However, this implies it is highly dependent on the method used (TRF vs. qPCR). In addition, some of the relationships are not fully convincing, such as those between food insecurity and fat scores and food consumption (as the authors acknowledged). Although the statistical approach is sound and sample sizes are adequate, some

analyses should be revised to clarify their findings. Overall, their work is interesting and may have important implications as it points towards telomere dynamics as a candidate mechanism to underlie life-history trade-offs. In general, the paper is well-written. Nonetheless, I would like to see a more elaborated context showing current evidence on the topic across birds, and not only - or mostly- in the study species and humans.

Specific comments

L 35. I miss a broader general context in the first paragraphs of the introduction. For example, the first sentence could suggest that the generality of this phenomenon is limited to very few bird species. I would prefer to see how general this process is and what is the current evidence to support -or not- this response across taxa, or at least, across bird species. For example, is fat storage/mass gain more likely to occur in short-lived passerines than in long-lived large-sized birds (e.g. raptors, seabirds...)? And what is more, is increased energy storage at the expense of investment in somatic maintenance/repair equally beneficial/adaptive for short- and long-lived species? It would be interesting to develop these ideas in the introduction/discussion sections.

L 38-49. Here is the opposite. Although interesting, the jump from passerines (mostly starlings) to humans seems illogical, especially given the differences in methodology (in the case of humans, not only observational, but probably less accurate and highly biased data) and likely also in the drivers and underlying physiological mechanisms. It seems that you try to generalize patterns across vertebrates, but to what extent are they given the diversity of life-history strategies that exist in the animal kingdom? What is in between passerines and humans?

L 46. The qualifying term “relevant” is certainly unnecessary here. Please, remove.

L 48. Add the species' Latin name

L 95. Please, indicate how many birds in total were captured.

L 96. Sex assignment through visual assessment seems to be highly reliable. Please, provide some details (or references) on the specific traits used to do this, as they may be very useful for other researchers working with the same study species.

L 112. There is an extra “prior”

L 140. Fat scoring of birds before starting the experiment would have been desirable. Do you have any reason for not having done so? Although both mass and fat storage are correlated, you cannot assume evidence for a treatment effect without having proof of change in this variable. It may also be other mechanisms underlying patterns of fat mobilization and storage (e.g. sex-specific responses, diseases...).

In addition, measuring body size (not only body mass) would have been desirable, especially for the comparison of birds at the beginning of the experiment. In contrast to body mass, body size is not expected to change. However, the lack of differences in body mass could be related to birds' size and reporting this result without controlling for body size is not convincing. If data are available, I encourage authors to use it as a control variable.

L 149-150. I understand that measuring food consumption at the individual level under your experimental conditions is almost impossible, but the conversion to g per bird per day eliminates any type of variation between individuals, which is in considerable contrast to the other variables used in this study. I really appreciate the authors' transparency throughout the paper and this particular issue has been duly acknowledged.

L 157. Please, indicate the measurement error associated to your calipers.

L 196-97 Indicate whether CV values of the control sample and ICC across individuals are within the acceptable ranges. Surely they are, but please provide accepted values and references for that.

L 201. What is/are the response variable/s in the models of experimental effects? and what is the difference between these and models for mass, TL and feather regrowth, if any? Please, clarify

Sex should also be included in your models as a fixed factor. Sex-specific hormones, even in juvenile birds, may influence some physiological and immunological parameters, which may in turn influence trade-offs between different life-history traits. As an example, experimental implantation of testosterone in adult female starlings strongly suppressed tail feather regrowth after experimental plucking (De Ridder et al. 2002 *The Auk* 119). Even though sexes are quite balanced in your study design, you cannot completely rule out an effect of sex on your results without testing it.

L 217. Using a meta-analysis with data from a single study is uncommon and would need more justification. Please, explain why it is needed or what is the added value to your statistical approach? I guess it is not only a "life jacket" to reinforce otherwise weaker evidence... why not to do such an analysis with data from different studies?

L 228. This is a bit intriguing. Although both groups follow a quite similar trend, the mass difference by insecurity status at week 14 seems to be motivated by a substantial decrease in body mass of control birds from week 8th, which then recovers from week 14 to 17, and drops onwards. How do you explain these changes in body mass within the control group and, in particular, the sharp decrease at week 14 (when in addition, the difference in TL is the smallest)?

L 238. Please, indicate what the lines and whiskers in the box plots represent.

L 274. How do you explain the shorter TL in the insecure birds at percentile 10?

L 295. Confidence intervals for Mass and aTL contain zero. Although this does not certainly mean that there is no treatment effect, it does imply that it is uncertain whether there is a treatment effect. The combined effect for both mass and fat score clarifies this issue, but I have serious doubts about whether it is appropriate to include fat score in this analysis (for the reasons explained above). I suggest authors to re-run the analysis without this variable.

L 314-319 Why not to perform the meta-analysis with data from all these studies instead?

L 317. Indicate (briefly) what are the main differences between the two methods of inducing food insecurity, as these can influence the derived results and conclusions.

L 353-354. Although transparency and sincerity are highly appreciated, it is a bit strange to insist on this idea when the main hypothesis of this study includes a specific prediction about this component of the trade-off. This is not only because the TRF is a high-precision method and you have great expertise in telomere dynamics, which is great, but rather because telomere dynamics may be a candidate mechanism to underlie life-history trade-offs. Of course there are other measures that may also be involved in the trade-offs associated with energy limitations that are worth discussing. I suggest rewriting this paragraph to discuss in more detail how DNA damage, immune function, stress hormones and any other physiological trait may be affected by food insecurity, and how these plastic or adaptive complex responses may help organisms to cope with natural variation in environmental conditions, including food availability.

In addition, you may want to discuss the potential impact of other factors naturally affecting wild birds, such as infection by haemosporidians on your response variables. Your particular study species is not the best candidate, as they usually have very low prevalence of infection by these parasites, but considering this in future studies with other bird species would be important. Avian malaria infection may or may not have direct short-term costs (e.g. decreases in body mass) but may affect body condition and immune responses of birds (e.g., Valkiūnas et al. 2006 *J. Parasitol.*, 92; Martínez-de la Puente et al. 2010 *Biol. Lett.*, 6; Marzal et al. 2008 *J. Evol. Biol.*, 21; Navarro et al. 2003 *Oikos*, 101) as well as susceptibility to oxidative stress (van de Crommenacker et al. 2002 *Proc. R. Soc. B* 279). On the other hand, diet quality /quantity may increase susceptibility to infectious diseases (Cornet et al. 2014 *J. Anim. Ecol.*, 83). All these effects may in turn influence telomere length and dynamics. For example, a causal relationship between avian malaria and faster telomere shortening has been demonstrated by the finding that telomeres shortened faster in experimentally infected captive birds compared with controls (Ashgar et al. 2015 *Science* 347). To add even more complexity, effects of infection by haemosporidians on telomere dynamics may be sex-specific (Sudyka et al. 2019 *The Science of Nature* 106). I believe it's worth considering these potential relationships in further studies.

L 362. Or the opposite! Shorter telomeres driven by poor environmental conditions -in a broad sense- could lead to faster pace-of-life, where investment in self-maintenance is reduced in order to save energy for reproduction, at the cost of somatic durability (see for instance Giradeau et al. 2019 *BioEssays*). You may want to further discuss these ideas and whether these patterns may also differ between individuals of different age or breeding prospects.

L 372 and 377. Please, provide Latin names.

L 386-391. Move this information to results.

L 392. I believe your results do not confirm, but rather suggest this may happen; evidence for fat scores and food intake is weak.

Hope you find these comments useful.

Reviewer: 2

Comments to the Author(s)

(See attached file, "RSOS-211099_review.pdf"): This study aimed at understanding the effects of food insecurity on energy storage and somatic maintenance. To do so, the authors carried out a 19-week experiment on 69 European starlings kept on captivity. They predicted that insecurity would increase energy storage (here measured as mass and fat deposition) but, as a trade-off, decrease somatic maintenance (here measured as telomere length and feather growth). The goal of the study is very clear and the justifications for the hypothesis, the experimental design and the predictions explicit and concise. Globally, the results of the study go in the direction of the predictions even though the effect of insecurity on mass and average telomere length was significant at some time points only. Nonetheless, the results remain convincing. More importantly, the discussion clearly highlights the significant and non-significant effects of the treatment, and the conclusions are justified by the results. The authors also highlight the limitations of the study.

My main comment is about the result section. I find misleading the description of the differences between treatments of the mean values of the different traits. I suggest to only keep the interpretation of the outputs of the statistical models (see comments 9, 12, 13 and 14 below).

1. Line 95: The authors should explain why the study focusses on juveniles only. Was it to standardise the age, and, therefore, initial telomere length?
2. Line 109: I understand that the authors want to highlight in this paragraph how the study dealt with ethical aspects. However, this sentence looks a bit out of place. I would move this sentence line 136, when explaining how birds were weighed.
3. Line 112: Delete one of the "prior".
4. Line 122: I would precise between brackets that "one compartment was open every 5 hours".
5. Line 123: For clarity, I would rephrase as "Thus, although the compartments were revealed at the same times as for the insecure treatment, the compartments always contained food".
6. Line 127: This sentence suggests that the compartments do not open at fix interval; please clarify.
7. Line 130: This sentence suggests that birds always had access to food, even in the insecure treatment. This would mean that, contrary to what is written in the introduction, the manipulation did not introduce restriction of food access. Please clarify.
8. Line 136: I would move this sentence ("Body masses [...]") after "[...] 19 weeks of treatment." and remove the part about fat score as the fact that the experimenter was not blind to treatment when measuring fat deposition is written further in the paragraph.
9. Line 227: I would remove this sentence ("Insecure birds were [...]") because, as written after, insecure birds were significantly heavier than control birds at weeks 14 and 19 only.
10. Figure 1: I suggest adding letters to the 3 different graphs to highlight what are the significant differences.
11. Line 245: The model should be described in Statistical analyses.
12. Line 248: Because the effect of insecurity is not significant, this sentence is misleading. Alternatively, the authors may describe the marginally non-significant interaction between week and insecurity.
13. Line 263: Again, I think that interpreting the differences between the means (while ignoring the variance) is wrong. The interpretation should be only made from the output of the statistical models. This sentence says that insecure birds had shorter aTL than control birds while the previous sentence says that the effect of the treatment is marginally non-significant. Alternatively, the authors may write that "while the mean value of aTL was lower in the insecure group than in the control group, this difference was only marginally non-significant".
14. Line 265: Both CI include 0; if I am correct, this means that aTL did not differ between the two groups whenever it was at week 2 or 14.
15. Line 271: It would be better to interpret the interaction between insecurity and percentile from pairwise comparisons (using the "eemans" function for instance).
16. Figure 2: Figure 2A suggests a rather strong effect of insecurity on average telomere length. However, this effect is statistically weak. Therefore, I think the figure is misleading and the

authors should represent aTL change from baseline rather than aTL (similarly to Figure 1A). The authors should also use letters or asterisks to highlight the differences that are significant.

17. Figure 2B: I find this figure very convincing. The interpretation of the interaction between insecurity and percentile should be emphasised using pairwise comparisons.

18. Line 282: It is probably worth repeating here when the feathers were plucked.

19. Lines 363-391: The two paragraphs about telomeres do not strictly relate to the study but provide interesting results that are valuable for research on telomeres in general.

Reviewer: 3

Comments to the Author(s)

General comments:

Andrews and colleagues conducted a study to investigate how food insecurity affect energy storage and somatic maintenance in European starlings. The authors found that food insecurity resulted in increased body mass and fat score, as well as decreased telomere length and slower feather regeneration. This is a well-designed study and a well-written manuscript. I only have a few general comments and some minor suggestions to improve this manuscript. The manuscript would benefit from addition of a figure with experimental timeline, as well as pictures or videos of the automated feeder and different phases of feather regeneration after plucking. Furthermore, I suggest discussing more about the validity of erythrocyte telomere length (more below) and the role of telomerase in the context of food security and somatic maintenance. Specific comments are detailed below.

Specific comments:

Line 48: Please include Latin name of zebra finches.

Line 52: Please include Latin name of European starlings.

Line 64-66: Erythrocytes are terminally differentiated cells. They should not be proliferating.

Line 65: Are there any evidence that shows that TL in erythrocytes is representative of TL in other tissues?

Line 79-82: I think this belongs to the methods section.

Line 108: Is 11-19 days sufficient for acclimation to captivity? Did body mass change throughout this period?

Line 110: What does "birds' dawn" mean?

Line 121: A picture, or even better, video of how the automated feeders work would be helpful.

Line 198: The study includes both males and females but 'sex' was not included in the analysis. Was there a sex-specific effect and/or interaction?

Figure 1A: It seems like there is a significant drop in body mass in both groups but especially in the control birds from week 17 to week 19. Why?

Line 382-391: I think the measurement of TL in avian erythrocytes warrants a bit more discussion. How does this extrapolate to TL of other somatic tissues? Although the authors did not measure telomerase activity, the role of telomerase should be discussed briefly in the context of this study.

===PREPARING YOUR MANUSCRIPT===

===PREPARING YOUR REVISION IN SCHOLARONE===

- If you are requesting a discretionary waiver for the article processing charge, the waiver form must be included at this step.
- If you are providing image files for potential cover images, please upload these at this step, and inform the editorial office you have done so. You must hold the copyright to any image provided.
- A copy of your point-by-point response to referees and Editors. This will expedite the preparation of your proof.

- Ensure that your data access statement meets the requirements at <https://royalsociety.org/journals/authors/author-guidelines/#data>. You should ensure that you cite the dataset in your reference list. If you have deposited data etc in the Dryad repository, please only include the 'For publication' link at this stage. You should remove the 'For review' link.
- If you are requesting an article processing charge waiver, you must select the relevant waiver option (if requesting a discretionary waiver, the form should have been uploaded at Step 3 'File upload' above).
- If you have uploaded ESM files, please ensure you follow the guidance at <https://royalsociety.org/journals/authors/author-guidelines/#supplementary-material> to include a suitable title and informative caption. An example of appropriate titling and captioning may be found at https://figshare.com/articles/Table_S2_from_Is_there_a_trade-off_between_peak_performance_and_performance_breadth_across_temperatures_for_aerobic_scope_in_teleost_fishes_/3843624.

Author's Response to Decision Letter for (RSOS-211099.R0)

See Appendix A.

Decision letter (RSOS-211099.R1)

Dear Dr Nettle,

I am pleased to inform you that your manuscript entitled "Exposure to food insecurity increases energy storage and reduces somatic maintenance in European starlings" is now accepted for publication in Royal Society Open Science.

Please ensure that you send to the editorial office individual files for each table included in your manuscript. You can send these in a zip folder if more convenient. Failure to provide these files may delay the processing of your proof.

on behalf of Dr Kimberley Mathot (Associate Editor) and Kevin Padian (Subject Editor)
openscience@royalsociety.org

Appendix A

Andrews et al.: Response to reviews

Thank you for submitting your manuscript for consideration at Royal Society Open Science. I have now received reports from three referees with relevant expertise, who were all generally very positive about the work. The reviews were very constructive and point out several areas where the manuscript could be improved/clarified. Please provide a point-by-point response to all the referee comments, including the following points which provide my own view on some of the more substantial comments from the referees.

Thank you for these thorough and timely reports. We have prepared a revised version that we hope addresses them all satisfactorily, as detailed point by point below.

1) Both referee #1 and referee #3 raise questions about the possibility of sex specific effects. It seems reasonable to expect that males and females may respond to food insecurity differently. For example, loss of fat stores may impact female reproductive potential more than it would in males, and that females might be more willing to sacrifice somatic maintenance for fat stores. From your introduction (lines 38-39), it seems like this is the observed pattern in humans; in females, but not males, food insecurity is associated with higher body mass index. Perhaps my interpretation of this is incorrect, and the studies only involved female subjects, hence the qualifier. Could you please clarify whether these previous studies showed sex-specific effects, and if so, why sex specific effects weren't considered here? I believe with your sample sizes, you would have the power to address the interaction between food insecurity and sex.

Thanks for this. Studies of humans do indeed find sex-specific effects (we have reworded the introduction to make this clear). Previous studies on starlings have not found them, which is why we did not explore the possibility. However, our study is better powered than previous ones to investigate this possibility. We have now rerun all the analyses including sex, in interaction with treatment, in the model. There are no interactions involving sex, and no main effects of sex other than the expected one on mass (male starlings are larger and heavier than females). For parsimony we therefore stick with the simpler models (no sex) in the main results section, but state that we have also run the models including interactions with sex and find none. This is stated at the end of the data analysis section. The R scripts published along with the paper include the analyses with and without sex included in the models.

2) Referee # 1 asks for clarification about the use of meta-analysis on multiple response variables from the same study and suggests the possibility of including previously published effect sizes in the current analysis. While I agree that such an expanded meta-analysis would be very timely and insightful, I think this is beyond the scope of the current work. Instead, can you please just provide a brief rationale of the approach you have used here.

Thank you. We have provided such a justification in Data Analysis (see response to reviewer 1), but we agree that combining the effects from different studies would be beyond the present scope, as well as answering a slightly different question. We have also included an additional method of making inferences based on the set of measures of energy storage, and the set of measures of somatic investment, namely the inverse normal method of combining p-values (see response to reviewer 1). This method points the same way as the meta-analysis.

3) Referee #2 had some concerns about the use of the term significant to describe effects throughout the manuscript when for certain specific time points, the effects were not statistically significant at a level of $\alpha \leq 0.05$. I believe this concern might be resolved by i) consistently relying on effect sizes and confidence intervals for drawing inference about effects and ii) providing an explicit statement about how you interpret support for a given effect (e.g., combination of estimated effect size and the likelihood that the effect is different from zero to infer strong, moderate, weak or no support). Referee #2 also felt that the results presented in Figure 2A were not consistent with the

results from your statistical analyses, which gave a $p = 0.08$. To my eyes, they do appear consistent as the lack of difference in week 14 could be sufficient to make the overall effect “not significant” -but please double check.

We are happy that the statements we make as regards figure 2A are correct and consistent. We have also amended some wording in response to referee 2's suggestions. We are happy that we are using the term 'significant' appropriately (i.e. iff $p < 0.05$). We prefer not to do many multiple statistical tests (e.g. one at each time point), and make too much of the 'significance' of each one on its own. It is much sounder statistically to fit one model to all the data, and then interpret the significance of the main effect of treatment, and the interaction of treatment with time point. This is what we do. When we discuss individual time points, we concentrate on the effect size of the difference and its confidence interval, and do not provide separate p-values.

Reviewer: 1

L 35. I miss a broader general context in the first paragraphs of the introduction. For example, the first sentence could suggest that the generality of this phenomenon is limited to very few bird species. I would prefer to see how general this process is and what is the current evidence to support –or not- this response across taxa, or at least, across bird species. For example, is fat storage/mass gain more likely to occur in short-lived passerines than in long-lived large-sized birds (e.g. raptors, seabirds...)? And what is more, is increased energy storage at the expense of investment in somatic maintenance/repair equally beneficial/adaptive for short- and long-lived species? It would be interesting to develop these ideas in the introduction/discussion sections.

We agree that these are interesting questions. However, the truth is that little is currently known on the questions above. Moreover, nothing we actually study in our experiment (which is on one passerine bird) could possibly answer these questions. We would like to write a review article in future that would sketch out some of these issues, but here, in the interest of brevity, we would like to focus on the species we study in this paper and the questions we can hope to answer here.

L 38-49. Here is the opposite. Although interesting, the jump from passerines (mostly starlings) to humans seems illogical, especially given the differences in methodology (in the case of humans, not only observational, but probably less accurate and highly biased data) and likely also in the drivers and underlying physiological mechanisms. It seems that you try to generalize patterns across vertebrates, but to what extent are they given the diversity of life-history strategies that exist in the animal kingdom? What is in between passerines and humans?

Good point. There are really only three groups of species that have been well-studied in this regard: small passerines, especially starlings; rodents; and humans. Hence our abrupt leap. We have added citations to the rodent work and also rephrased to make the leap seem less abrupt.

L 46. The qualifying term “relevant” is certainly unnecessary here. Please, remove.

Removed

L 48. Add the species' Latin name

Done

L 95. Please, indicate how many birds in total were captured.

Non-juvenile birds were released immediately on removal from the net, and unfortunately we have no record of how many of those there were. Our recollection is that the majority were juveniles. We have clarified the wording to make clear that the 70 refers to the number of juveniles retained, not the number of birds netted.

L 96. Sex assignment through visual assessment seems to be highly reliable. Please, provide some

details (or references) on the specific traits used to do this, as they may be very useful for other researchers working with the same study species.

Indeed, in juveniles the principal cue is iris colour. We have stated this and added a reference that shows the reliability of this method against karyotype.

L 112. There is an extra “prior”

Thanks, corrected.

L 140. Fat scoring of birds before starting the experiment would have been desirable. Do you have any reason for not having done so? Although both mass and fat storage are correlated, you cannot assume evidence for a treatment effect without having proof of change in this variable. It may also be other mechanisms underling patterns of fat mobilization and storage (e.g sex-specific responses, diseases...).

It is indeed a shame that we did not fat score at capture; this was an oversight. However, we do not agree that we have no evidence for a treatment effect on fat score without a pre-treatment measure. In many experiments, the only measurement of the dependent variable is post-treatment. Assignment to treatments was random, and so it is a reasonable inference that any systematic difference in fat score between the two groups after the treatment is due to the effects of the treatment.

In addition, measuring body size (not only body mass) would have been desirable, especially for the comparison of birds at the beginning of the experiment. In contrast to body mass, body size is not expected to change. However, the lack of differences in body mass could be related to birds' size and reporting this result without controlling for body size is not convincing. If data are available, I encourage authors to use it as a control variable.

Good point. We did in fact measure skeletal size via tarsus length at capture. We have now included this as a covariate in the models for mass. We agree this makes the analyses more convincing. The conclusions are unaffected.

L 149-150. I understand that measuring food consumption at the individual level under your experimental conditions is almost impossible, but the conversion to g per bird per day eliminates any type of variation between individuals, which is in considerable contrast to the other variables used in this study. I really appreciate the authors' transparency throughout the paper and this particular issue has been duly acknowledged.

Thanks – we take this comment to mean no change is required. We agree that this is an important limitation but there was no practical way to overcome it within our experimental setup.

L 157. Please, indicate the measurement error associated to your calipers.

0.1 mm, added.

L 196-97 Indicate whether CV values of the control sample and ICC across individuals are within the acceptable ranges. Surely they are, but please provide accepted values and references for that.

We have added a reference to a recent review on reproducibility and repeatability in TL studies. Our ICC value of 0.77 is acceptable, though we note that the true value is likely to be higher, since TL was genuinely varying within individuals over the course of the study.

L 201. What is/are the response variable/s in the models of experimental effects? and what is the difference between these and models for mass, TL and feather regrowth, if any? Please, clarify

We have rephrased as our wording here was evidently confusing. The models of experimental effects were the same ones as those for mass, TL and feather regrowth. There was a separate model for each of the response variables (mass, fat score, etc.). We have rephrased here for clarity.

Sex should also be included in your models as a fixed factor. Sex-specific hormones, even in juvenile birds, may influence some physiological and immunological parameters, which may in turn influence trade-offs between different life-history traits. As an example, experimental implantation of testosterone in adult female starlings strongly suppressed tail feather regrowth after experimental plucking (De Ridder et al. 2002 The Auk 119). Even though sexes are quite balanced in your study design, you cannot completely rule out an effect of sex on your results without testing it.

As mentioned in the comments to the editor, we have rerun all analyses additionally including sex as suggested above. There are no significant interactions between sex and treatment, and the only significant main effect of sex is the expected one on mass. No other conclusions are altered by the inclusion of sex. We now explain this at the end of the data analysis section.

L 217. Using a meta-analysis with data from a single study is uncommon and would need more justification. Please, explain why it is needed or what is the added value to your statistical approach? I guess it is not only a “life jacket” to reinforce otherwise weaker evidence... why not to do such an analysis with data from different studies?

In this study, we have multiple variables that are indicators of the same underlying process (i.e. mass and fat score are both indicators of energy storage, telomere length and feather regrowth are both indicators of maintenance). A traditional approach here would be MANOVA, which would ask whether the experimental treatment had a significant effect on the set of dependent variables. A MANOVA test would often be significant even if the effects on the individual variables were marginal. However, we cannot do MANOVA here, since the structure of the data is different from the different variables (fat score is measured just once, whereas mass is measured multiple times; telomere length is not measured the same number of times as feather regrowth). The alternative possibilities are therefore meta-analysis of the standardized effects (which we used in the first version), or the inverse normal method for combining p-values (see Zaykin D V. (2011) Optimally weighted Z-test is a powerful method for combining probabilities in meta-analysis. *J. Evol. Biol.* **24**, 1836–1841). We have now added this second method. Both of these methods lead to the same conclusion in this case. We have added a sentence justifying why we do this rather than MANOVA in the data analysis section. In addition, we have also added the inverse normal method of combined p-value calculation as well.

As for including effects from different studies, that would be a substantial project in its own right, beyond our current scope. Besides, no other study has exactly our combination of effects.

L 228. This is a bit intriguing. Although both groups follow a quite similar trend, the mass difference by insecurity status at week 14 seems to be motivated by a substantial decrease in body mass of control birds from week 8th, which then recovers from week 14 to 17, and drops onwards. How do you explain these changes in body mass within the control group and, in particular, the sharp decrease at week 14 (when in addition, the difference in TL is the smallest)?

Starling mass is quite variable over days and weeks in captivity. For example, changes in ambient temperature can lead to a gain or loss of several grams. Also, the birds may have been exhibiting long-term adaptation to captive conditions (the initial gain of mass during the experiment, followed by a decline, in the control group). Thus, we can really only interpret the *difference* between the two groups, for whom all these factors should have been the same. We have no clear explanation of the temporal trend in the control group. We do note in Data Analysis that there are non-linear trends in mass due to factors unrelated to treatment, and hence treatment week needs to be included in the models as a factor.

L 238. Please, indicate what the lines and whiskers in the box plots represent.

The boxes represent the quartiles and medians, and we have added this information. The whiskers

are complex but there is a standard convention: they represent the upper/lower quartile + or - 1.5 * the inter-quartile range, or else the minimum/maximum of the data if more central. We have not added this but it is the default R settings, and the default convention for boxplots.

L 274. How do you explain the shorter TL in the insecure birds at percentile 10?

We don't have any clear explanation for this: we would note that the confidence interval for this difference includes zero.

L 295. Confidence intervals for Mass and aTL contain zero. Although this does not certainly mean that there is no treatment effect, it does imply that it is uncertain whether there is a treatment effect. The combined effect for both mass and fat score clarifies this issue, but I have serious doubts about whether it is appropriate to include fat score in this analysis (for the reasons explained above). I suggest authors to re-run the analysis without this variable.

We agree that the confidence intervals for mass and aTL contain zero, and hence that it is not certain there is a treatment effect. We are quite explicit about this. But, we have multiple indicators of each of the biological quantities we were attempting to estimate (fat score as well as mass for energy storage), and this is why it seemed legitimate to us to combine the evidence from the two indicators in a principled way. This is what we do with the combined p-value, and the meta-analysis. We are unclear why the reviewer is so dubious about the fat score variable: the groups were randomly assigned, and so there is no reason to expect they would have differed at baseline. Thus, finding that they differ post-treatment seems to us to be a legitimate finding. For many, perhaps most, experimental effects, there is only a post-treatment measurement of the dependent variable.

L 314-319 Why not to perform the meta-analysis with data from all these studies instead?

We agree, and indeed we are involved in a slow project to perform such a meta-analysis, but that would be a different project from the current paper.

L 317. Indicate (briefly) what are the main differences between the two methods of inducing food insecurity, as these can influence the derived results and conclusions.

We have added a sentence explaining the different protocols.

L 353-354. Although transparency and sincerity are highly appreciated, it is a bit strange to insist on this idea when the main hypothesis of this study includes a specific prediction about this component of the trade-off. This is not only because the TRF is a high-precision method and you have great expertise in telomere dynamics, which is great, but rather because telomere dynamics may be a candidate mechanism to underlie life-history trade-offs. Of course there are other measures that may also be involved in the trade-offs associated with energy limitations that are worth discussing. I suggest rewriting this paragraph to discuss in more detail how DNA damage, immune function, stress hormones and any other physiological trait may be affected by food insecurity, and how these plastic or adaptive complex responses may help organisms to cope with natural variation in environmental conditions, including food availability.

We have deleted the sentence about our choice of measures being opportunistic rather than principled, but the point remains that there are other possible measures of somatic investment, and hence either we got lucky with the ones we chose, or food insecurity has effects across several of them. I don't think any of us think of telomere dynamics as a candidate mechanism underlying these trade-offs. Telomeres behave more like a marker of processes in which they are not actually causally important, than a mechanism. Certainly, we have no reason to argue that telomeres are a causal important mechanism in orchestrating these trade-offs: more like a casualty when the trade-offs are made.

In addition, you may want to discuss the potential impact of other factors naturally affecting wild birds, such as infection by haemosporidians on your response variables. Your particular study species is not the best candidate, as they usually have very low prevalence of infection by these parasites, but considering this in future studies with other bird species would be important. Avian malaria infection may or may not have direct short-term costs (e.g. decreases in body mass) but may affect body condition and immune responses of birds (e.g., Valkiūnas et al. 2006 J. Parasitol., 92; Martínez-de la Puente et al. 2010 Biol. Lett., 6; Marzal et al. 2008 J. Evol. Biol., 21; Navarro et al. 2003 Oikos, 101) as well as susceptibility to oxidative stress (van de Crommenacker et al. 2002 Proc. R. Soc. B 279). On the other hand, diet quality /quantity may increase susceptibility to infectious diseases (Cornet et al. 2014 J. Anim. Ecol., 83). All these effects may in turn influence telomere length and dynamics. For example, a causal relationship between avian malaria and faster telomere shortening has been demonstrated by the finding that telomeres shortened faster in experimentally infected captive birds compared with controls (Ashgar et al. 2015 Science 347). To add even more complexity, effects of infection by haemosporidians on telomere dynamics may be sex-specific (Sudyka et al. 2019 The Science of Nature 106). I believe it's worth considering these potential relationships in further studies. We agree that it is important to study these relationships and effects in future studies. However, with random assignment of wild birds to our experimental conditions, these kinds of factors can only constitute 'noise' in respect of our present experiment and aims.

L 362. Or the opposite! Shorter telomeres driven by poor environmental conditions –in a broad sense– could lead to faster pace-of-life, where investment in self-maintenance is reduced in order to save energy for reproduction, at the cost of somatic durability (see for instance Giradeau et al. 2019 BioEssays). You may want to further discuss these ideas and whether these patterns may also differ between individuals of different age or breeding prospects.

This is a good point. We are sceptical of the telomere-messenger hypothesis (the hypothesis that telomeres play an important orchestrating role in life-history trade-offs), but it is true that food insecurity could be expected to either increase or decrease reproductive investment, depending on how individuals trade-off survival and reproductive effort. Without wanting to go into this, we have amended the word 'reduced' to 'altered', to reflect at least the theoretical possibility of effects in either direction.

L 372 and 377. Please, provide Latin names.
Done.

L 386-391. Move this information to results.
Done

L 392. I believe your results do not confirm, but rather suggest this may happen; evidence for fat scores and food intake is weak.
We have weakened the wording to 'suggest'.

Reviewer: 2

My main comment is about the result section. I find misleading the description of the differences between treatments of the mean values of the different traits. I suggest to only keep the interpretation of the outputs of the statistical models (see comments 9, 12, 13 and 14 below). Please see responses to those individual comments.

1. Line 95: The authors should explain why the study focusses on juveniles only. Was it to standardise the age, and, therefore, initial telomere length?

The principal reason was to standardise age within the sample. Juvenile status is very obvious and ensures all the subjects hatched in the same year. In addition, it has the advantage that although TL will still be very variable for genetic reasons, it will at least be changing rapidly that early in life, maximizing the chances of detecting attrition. We have added a justification sentence to the methods

2. Line 109: I understand that the authors want to highlight in this paragraph how the study dealt with ethical aspects. However, this sentence looks a bit out of place. I would move this sentence line 136, when explaining how birds were weighed.

We agree, we have cut this sentence.

3. Line 112: Delete one of the "prior".

Thanks, corrected.

4. Line 122: I would precise between brackets that "one compartment was open every 5 hours".

We have added such a parenthesis.

5. Line 123: For clarity, I would rephrase as "Thus, although the compartments were revealed at the same times as for the insecure treatment, the compartments always contained food".

We have rephrased slightly. It is surprising how hard it is to state what appears such a simple thing.

6. Line 127: This sentence suggests that the compartments do not open at fix interval; please clarify.

That is correct, they do not open at fixed intervals. Instead, the time of opening can be programmed. We reprogrammed the feeders every day. We have added a clause to make this clearer.

7. Line 130: This sentence suggests that birds always had access to food, even in the insecure treatment. This would mean that, contrary to what is written in the introduction, the manipulation did not introduce restriction of food access. Please clarify.

We have rephrased slightly to try to make this clearer. In the insecure treatment, they had an empty compartment for 5 hours a day, but when they had a non-empty compartment (i.e. the remaining 19 hours), the food was so plentiful that they could eat as much as they wanted. In other words, we ensured that food never ran out *other* than in the intended 5 h of deprivation.

8. Line 136: I would move this sentence ("Body masses [...]") after "[...] 19 weeks of treatment." and remove the part about fat score as the fact that the experimenter was not blind to treatment when measuring fat deposition is written further in the paragraph.

Thanks, amended.

9. Line 227: I would remove this sentence ("Insecure birds were [...]") because, as written after, insecure birds were significantly heavier than control birds at weeks 14 and 19 only.

Ok, we have rephrased to: 'Figure 1A shows mean mass by treatment group across time'. The graph shows that they were heavier, just not significantly so at all times.

10. Figure 1: I suggest adding letters to the 3 different graphs to highlight what are the significant differences.

We don't really believe that multiple t-tests on all the different times points, with interpretation based on the 'significance' of each separate test, is statistically appropriate, although this is often done in biomedical papers with multiple time points. Instead it is better to fit a single model to all the data, and base inference on the main effect of treatment and the interaction between treatment and time. Thus, we would prefer not to follow this advice as it would make the reader focus too

much on the difference of significance between different time points, rather than the overall interaction between treatment and time.

11. Line 245: *The model should be described in Statistical analyses.*

We just wanted to remind the reader what the predictors were without them having to look back to the methods section. We have abbreviated this sentence.

12. Line 248: *Because the effect of insecurity is not significant, this sentence is misleading.*

Alternatively, the authors may describe the marginally non-significant interaction between week and insecurity.

We just wanted to make clear that to the extent there was any difference, it was going in the opposite direction to the prediction. We have rephrased to 'The non-significant trend was for insecure birds to consume slightly less overall' so there is no ambiguity that we are not claiming a significant difference here.

13. Line 263: *Again, I think that interpreting the differences between the means (while ignoring the variance) is wrong. The interpretation should be only made from the output of the statistical models. This sentence says that insecure birds had shorter aTL than control birds while the previous sentence says that the effect of the treatment is marginally non-significant. Alternatively, the authors may write that "while the mean value of aTL was lower in the insecure group than in the control group, this difference was only marginally non-significant".*

This is fine – we have amended the wording. I suppose we were assuming that one can say that the mean in group A is less than the mean in group B as a *descriptive* statement, without needing to imply that it is true as an *inference*, i.e. that it is a significant difference. Anyway, we have rephrased.

14. Line 265: *Both CI include 0; if I am correct, this means that aTL did not differ between the two groups whenever it was at week 2 or 14.*

Again, we are not keen to base inference on multiple t-tests at every time point – it is sounder to evaluate an overall model with a main effect of treatment and an interaction between treatment and time.

15. Line 271: *It would be better to interpret the interaction between insecurity and percentile from pairwise comparisons (using the "emmeans" function for instance).*

Figure 2B is generated using the estimated marginal means (and their standard errors) from the statistical model, using the emmeans package, and so in effect that is what we are doing.

16. Figure 2: *Figure 2A suggests a rather strong effect of insecurity on average telomere length. However, this effect is statistically weak. Therefore, I think the figure is misleading and the authors should represent aTL change from baseline rather than aTL (similarly to Figure 1A). The authors should also use letters or asterisks to highlight the differences that are significant.*

Using aTL change rather than aTL looks very similar to the present figure 2A. We hope the figure is not misleading – it comes straight from the data. It is compatible with the main effect of treatment not being significant because (a) the error bars are one standard error, not 95% confidence intervals; and (b) they overlap at one of the time points. We also disagree philosophically about a lot of separate inferences about statistical significance at different time points – it is better to ask, in a single model, is there a main effect of treatment, and does it interact with time point.

17. Figure 2B: *I find this figure very convincing. The interpretation of the interaction between insecurity and percentile should be emphasised using pairwise comparisons.*

Again, we are not keen on stressing the 'difference in significance' across percentiles, when what matters is that there is a significant interaction overall.

18. Line 282: *It is probably worth repeating here when the feathers were plucked.*
We have added this information.

19. Lines 363-391: *The two paragraphs about telomeres do not strictly relate to the study but provide interesting results that are valuable for research on telomeres in general.*

This is to our knowledge the first time anyone has reported the base pairs length of telomeres in starling (our previous studies used qPCR and hence report only relative telomere length), which is why we spent the time discussing this information here.

Reviewer: 3

Line 48: *Please include Latin name of zebra finches.*

Done

Line 52: *Please include Latin name of European starlings.*

Done

Line 64-66: *Erythrocytes are terminally differentiated cells. They should not be proliferating.*

Individual erythrocytes are terminally differentiated. However, the tissue as a whole has a high rate of cell replacement, and the haematopoietic stem cells from which erythrocytes derive proliferate rapidly. Hence, TL is more dynamic in erythrocytes than, say, muscle. Calling blood a proliferative tissue is standard in the field (though we have changed 'erythrocytes' to 'blood' here to reflect that individual erythrocytes do not proliferate).

Line 65: *Are there any evidence that shows that TL in erythrocytes is representative of TL in other tissues?*

None from starlings, though there is a good study in zebra finches, which we now cite, showing that TL from erythrocytes is a good proxy for TL in other tissues, even though the absolute lengths are different. The same is true in humans (though leucocytes rather than erythrocytes).

Line 79-82: *I think this belongs to the methods section.*

With respect, we would rather leave this in the Introduction, for two reasons: (a) it is a distinctive feature of our study to have measured starling telomeres with TRF. For example, it allows us to report actual TL in base pairs, whereas previous studies have only been able to capture relative TL; (b) because TRF gives a *distribution* of TLs, not just an average, we were actually able to ask different questions by using TRF. We feel these are sufficiently central design features of our study to be mentioned in the Introduction.

Line 108: *Is 11-19 days sufficient for acclimation to captivity? Did body mass change throughout this period?*

Body mass is quite labile in starlings, and our experience with keeping them in captivity is that body mass never stops changing, for example with age, with the seasons, and with long term captivity. So we cannot show that after 11-19 days they are fully acclimated or that their body masses are stable. This was just a pragmatic period of time for them to at least get used to the feeders and the day length. There was no principled criterion for showing that they were 'ready' for the experiment.

Line 110: *What does "birds' dawn" mean?*

This sentence is cut anyway, but the birds' dawn was not at the external dawn. Remember they were living under artificial light. We had their dawn mid-morning rather than at the time of the external dawn.

Line 121: A picture, or even better, video of how the automated feeders work would be helpful.
We still have the feeders but alas no longer have any starlings with which to demonstrate the foraging. A static picture will not help much, since the feeder resembles nothing so much as a large covered bowl.

Line 198: The study includes both males and females but 'sex' was not included in the analysis. Was there a sex-specific effect and/or interaction?

As described in the response to the editor, we have rerun all analyses additionally including sex, and find no significant interactions between sex and treatment. We explain this at the end of the data analysis section of the methods.

Figure 1A: It seems like there is a significant drop in body mass in both groups but especially in the control birds from week 17 to week 19. Why?

We can only speculate. The control birds returned to something close to the weights we caught them at. Perhaps the initial response to captivity is to experience it as unpredictable and put on weight, but in control birds this gradually resolved. Alternatively, it could be a seasonal pattern. We do acknowledge in the paper that mass changes over time for all kinds of non-experimental reasons, which is why it is important to include time point in the models.

Line 382-391: I think the measurement of TL in avian erythrocytes warrants a bit more discussion.

How does this extrapolate to TL of other somatic tissues? Although the authors did not measure telomerase activity, the role of telomerase should be discussed briefly in the context of this study.

We have added a brief paragraph making the point that since TL correlates across tissues and all tissues seem to have similar telomere shortening rates in adulthood, it is reasonable to assume that if food insecurity affects erythrocyte TL, it could affect TL in other tissues also.